# Can we collect health-related quality of life information from anticoagulated atrial fibrillation participants who have recently experienced a bleed? An observational feasibility study in primary and secondary care in Wales and through a UK online forum

Hayley Anne Hutchings [ID],[1] Kirsty J Lanyon [ID],[1] Gail Holland,[1] Raza Alikhan,[2] Rhys Jenkins,[3] Hamish Laing [ID],[4] Arfon Hughes,[5] Trudie Lobban,[6] Kevin Pollock,[7] Daniel Tod,[1] Steven Lister[7]

For numbered affiliations see end of article.

**Correspondence to**
Professor Hayley Anne Hutchings;
h.a.hutchings@swansea.ac.uk

## ABSTRACT

**Objective** To evaluate the feasibility of recruiting participants diagnosed with atrial fibrillation (AF) taking oral anticoagulation therapies (OATs) and recently experiencing a bleed to collect health-related quality of life (HRQoL) information.

**Design** Observational feasibility study. The study aimed to determine the feasibility of recruiting participants with minor and major bleeds, the most appropriate route for recruitment and the appropriateness of the patient-reported outcome measures (PROMs) selected for collecting HRQoL information in AF patients, and the preferred format of the surveys.

**Setting** Primary care, secondary care and via an online patient forum.

**Participants** The study population was adult patients (≥18) with AF taking OATs who had experienced a recent major or minor bleed within the last 4 weeks.

**Primary and secondary outcome measures** *Primary outcomes* – PROMs: EuroQol 5 Dimensions-5 Levels, Perception of Anticoagulant Treatment Questionnaire, part 2 only (part 2), atrial fibrillation effect on quality of life. *Secondary outcomes* – Location of bleed, bleed severity, current treatment, patient perceptions of HRQoL in relation to bleeding events.

**Results** We received initial expressions of interest from 103 participants. We subsequently recruited 32 participants to the study—14 from primary care and 18 through the AF forum. No participants were recruited through secondary care. Despite 32 participants consenting, only 26 initial surveys were completed. We received follow-up surveys from 11 participants (8 primary care and 3 AF forum). COVID-19 had a major impact on the study.

**Conclusions** Primary care was the most successful route for recruitment. Most participants recruited to the study experienced a minor bleed. Further ways to recruit in secondary care should be explored, especially to capture more serious bleeds.

## STRENGTHS AND LIMITATIONS OF THIS STUDY

⇒ Multiple routes were used to identify potential participants with atrial fibrillation (AF) who had experienced a recent bleed—primary care, secondary care and through an online AF forum.

⇒ Participants were able to complete the survey online (via REDCap), on a paper-based form or over the telephone.

⇒ Three validated patient-reported outcome measures (PROMs) were used to gather different aspects of the impact of bleeding on AF patients.

⇒ COVID-19 impacted on the recruitment of participants and alternative mechanisms for recruitment need to be considered.

**Trial registration number** The study is registered in the Clinicaltrials.gov database, NCT04921176.

## INTRODUCTION

Atrial fibrillation (AF) is the most common cardiac arrythmia (irregular heart rhythm disorder). Estimated overall prevalence has been reported to be between 0.4% and 1%.[1][2] UK estimates have suggested that AF prevalence is increasing and could be as high as 3.29%.[3] Many clinicians, however, debate the true global prevalence, believing that it is likely to be underestimated as many individuals are asymptomatic and are likely to go undiagnosed.[4] The rate increases with age over 60 years[5] and it is estimated to be much higher in individuals over 80 and in Western societies, with suggested rates as high as 14%.[6]

There are known to be high numbers of individuals with undiagnosed AF.[7–9]

AF is a chronic condition associated with significant risks of thromboembolism, stroke and mortality.[10 11] Oral anti-coagulation therapies (OATs)[1] are normally prescribed to reduce the risks of thromboembolism and AF-related stroke, but are associated with the side effect of bleeding, which in some cases can be serious.[10 12 13] These bleeding complications can include intracranial haemorrhage,[14] gastrointestinal bleeding and haematuria.[15] Bleeding as a side effect of OATs[16] remains a major challenge for clinicians that can often result in non-adherence and/ or resistance to therapy.[17] While hospitalised AF-related stroke rates have declined and are significantly associated with increased anticoagulant uptake,[18] the increased use of OATs has led to an increase in the number of bleeding events over time.[10]

Patients are treated by healthcare providers with the primary goal of improving their health and well-being. Historically this improvement in health has been judged by improvement in biochemical, histological, radiological or clinical assessments. It is hypothesised that this approach does not always correlate with improvement from the patient perspective.[19] From a patient perspective, improving health is reflected in the documentation of their symptoms and experience of healthcare provision, which are more appropriately collected directly from the patient.[20] With a move towards shared decision-making and patient-centred care, there is a growing recognition within the healthcare community of the importance of the patient perspective and the need to consider patient-reported outcome measures (PROMs) as a key component of a holistic approach to patient care.

PROMs were initially developed for research use, and many regulatory authorities such as the European Medicines Agency (EMA) and the USA Food and Drug Administration (FDA) advocate their use.[21–23] The US FDA defines a PROM as 'any report of the status of a patient's health condition that comes directly from the patient, without interpretation of the patient's response by a clinician or anyone else'.[22] The EMA state that 'Any outcome evaluated directly by the patient himself and based on the patient's perception of a disease and its treatment(s) is called a patient-reported outcome'.[24] PROMs include quality of life (QoL) measures, disease severity scales and patient experience measures.

The collection of PROMs aligns well with the increased drive within healthcare organisations for value-based healthcare, where organisations aim to achieve the best possible outcomes for patients with the available resources.[25 26] As more clinicians recognise the benefit of collecting PROMs in addition to measuring clinical outcomes, PROMs have seen an increased use in routine clinical practice.[27]

AF is thought to have a detrimental effect on health-related QoL (HRQoL) in patients.[28–33] This research has focused on exploring HRQoL following various treatments or procedures,[34–36] the development of AF specific HRQoL measures[37–40] or the use of existing HRQoL measures in AF populations.[41 42] There is some evidence that bleeding events may have an adverse effect on patient HRQoL[43 44] but there has been limited exploration of HRQoL following a bleed in AF patients.[42 45] The aim of this study was to determine the feasibility of collecting HRQoL information from anticoagulated AF participants who had recently experienced a bleed. If this proves feasible, we plan to embark on a fully powered study to evaluate the different dimensions of HRQoL of AF patients following a bleed.

## METHODS

### Selection of PROM to assess HRQoL

Prior to identifying sites and recruitment of participants, we undertook a process of PROM selection in order to select suitable tools that would allow us to assess HRQoL in AF participants who had experienced a recent bleed (reported elsewhere). This included a review of the literature to identify generic and disease specific PROMs that had been used in AF participants. Following the compilation of a list of possible PROMs, we undertook a rigorous evaluation of the identified PROMs that included an assessment of the overall validity of the PROMs, their content validity in terms of appropriateness of questions relating to bleeding events with AF, and how widely the PROM had previously been used in AF. This was informed by existing literature[46 47] and included a clear justification as to why the PROM was used.[48] In selecting the PROMs, we were also mindful to ensure the burden to participants was kept to a minimum, so also explored the number of questions within relevant PROMs. We worked with the two public and patient involvement representatives when assessing the PROMs for help in considering their usefulness to the study.

Following the PROM scoping process, we agreed on three PROMs for use in the study:

► EuroQol 5 Dimensions-5 Levels (EQ-5D-5L).[49]

The EQ-5D is made up of the EQ-5D UK crosswalk scores which ranges between −0.594 and 1.000. A higher index score indicates a better QoL. The EQ-5D VAS (Visual Analogue Scale) allows users to input a score of how good or bad their health is between 0 and 100, with a higher VAS score indicating a better QoL.

► Perception of Anticoagulant Treatment Questionnaire, Part 2 only (PACT-Q, Part 2).[50 51]

The PACT-Q Part 2 scores range from 0 to 100, for the convenience and satisfaction domains, with a higher score indicating higher convenience and higher satisfaction.

► Atrial fibrillation Effect on QualiTy-of-life (AFEQT).[41]

The disease-specific AFEQT global and specific domain scores range from 0 to 100. A score of 0 corresponds to complete disability, while a score of 100 corresponds to no disability.

## Participant recruitment

We recruited adult participants (aged ≥18 years) who had a diagnosis of AF and were actively prescribed OATs for their AF. We identified participants who had experienced a recent bleed, up to a maximum of 30 days prior to enrolment. No upper age limit was imposed, but participants had to fit all other inclusion and exclusion criteria.

### Inclusion criteria

► Adult participants (≥18 years).
► Participants who had a fluent understanding of English and were able to comprehend all study information and literature to provide fully informed consent.
► AF as the primary diagnosis.
► Had a major or minor bleed up to a maximum of 30 days prior to date of enrolment.
► Prescribed oral anticoagulation for AF.

### Exclusion criteria

► Pregnant women.
► Participants with active cancer.
► Participants unable to consent for themselves.
► Participants on concomitant antiplatelet therapy.

As the study was designed to test feasibility for a fully powered definitive study, we aimed to recruit between 50 and 80 participants in the first instance based on recommended sample size guidance.[52] Figure 1 outlines the stages of the study.

## Participating sites

For this feasibility study, we selected participants from the Swansea, UK area. We used three recruitment pathways for the study to test the feasibility of identifying and recruiting participants with minor and major bleeds:

► Primary care—Participants attending anticoagulation clinics of a large city general practice cluster made up of 8 practices and serving almost 51 000 patients in Swansea, UK. Participants were identified by practice nurse/pharmacist as having a recent bleed.
► Secondary care—Participants admitted to the emergency department of a large UK University teaching hospital in Swansea, UK with a bleed were identified by research nurses/secondary care-based general practitioner.
► Direct participant—The study was publicised on the Arrhythmia Alliance website and newsletters (Arrhythmia Alliance UK heartrhythmalliance.org) and individuals who formed part of this online forum and database were invited to participate.

When participants were identified through primary and secondary care, they were provided with a study information sheet for their consideration. If the patient was willing to participate, they were asked to return the expression of interest form to the study researcher in the supplied freepost envelope. The study researcher then contacted participants to provide them with a 'study pack'. As we aimed to determine the best approach for collecting the HRQoL information, we offered the participants the option of an electronic link to a REDCap survey, a paper survey sent to their home with a freepost envelope, or the option to complete the survey by telephone. Individuals from the AF forum were initially provided with a weblink to the REDCap survey through the website. Latterly, they were given the option to contact the researcher and be supplied with a paper survey.

## Participants survey pack

We collected basic demographics from participants, details of their current OAT, existing comorbidities and details of any bleed(s) they had experienced (ie, date, severity and bleed location). We asked all participants to complete three PROMs for the study survey at baseline: the EQ-5D-5L,[49] PACT-Q Part 2[50 51] and the AFEQT.[41]

We estimated that completion of all study questionnaires would take no longer than 30 min.

Participants who agreed to participate were asked to complete the survey at two time points—once following consent, and then again at 90±14 days postenrolment. Participants who completed the survey digitally via the REDCap system received a second online link to complete the survey; those who completed paper forms were sent a follow-up via post (along with a stamped addressed envelope for ease of return); and those who completed over the phone were contacted via telephone to complete the follow-up. Participants were contacted 3 weeks prior to the 90-day follow-up date. If no reply was received within 2 weeks of first correspondence, a reminder letter (with a second copy of all questionnaires) was sent, or a follow-up email or telephone call was made.

## Data analysis

Quantitative data were checked, cleaned and collated in REDCap prior to being transferred to Stata V.17.0. for analyses. As this was a feasibility study and not powered to detect differences between groups/OATs, no formal statistical comparisons were made. Data are presented descriptively as number, percentages or means (SD). HRQoL scores were calculated according to developers' guidance. Descriptive HRQoL scores are reported for AF participants following a minor or major bleed[53 54] while anticoagulated and by recruitment route. Categorisation of the bleed into minor or major was undertaken by a clinical professional (RA).

Participants were required to provide consent before enrolment into the study. Prior to consent, we provided participants with an information sheet which ensured that they were clearly and fully informed about the purpose of the study, potential risks, use of their data, and their rights and responsibilities when participating in the study. They were also told that they had the right to withdraw at any point in the study. Participants who completed the survey via the REDCap system consented to the study electronically. Participants completing paper forms were required to complete a paper consent and we obtained verbal consent for those completing over the telephone.

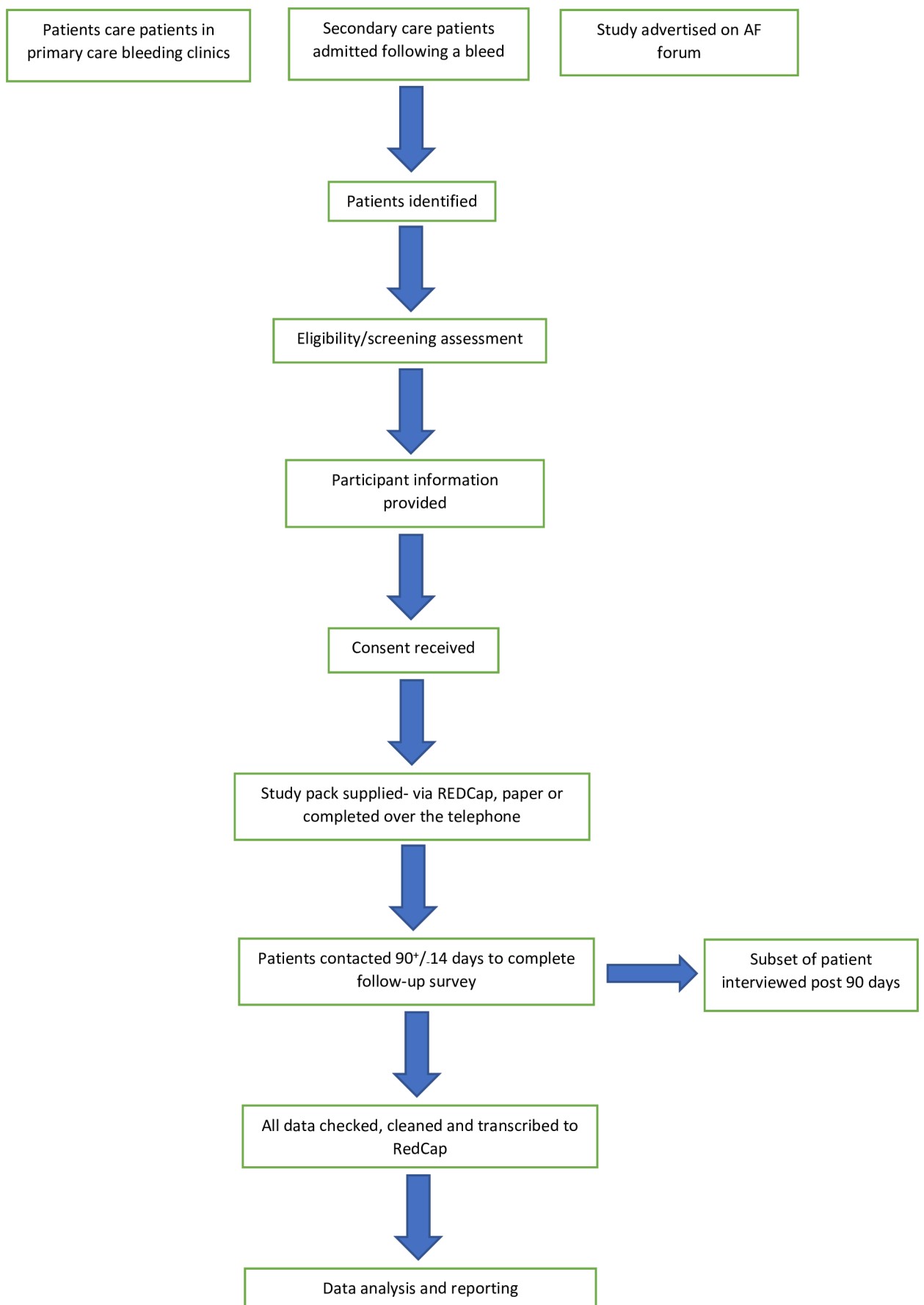

**Figure 1** Flow chart of the stages involved in the study. AF, atrial fibrillation.

We asked participants for informed consent for their anonymised data and interview quotes (where relevant) to be used in any future publications.

We were aware that participants may have experienced low mood or anxiety associated with their AF. If a participant reported, or showed signs of low mood, distress or anxiety, we encouraged them to discuss this with their primary or secondary care researcher and the participant was signposted to local relevant services or advised to contact their GP.

We stored all data securely within the password-protected REDCap secure web application while the study was conducted, and subsequently anonymised for analysis purposes.

### COVID implications/changes

This study was originally conceived and designed before the COVID-19 pandemic. Shortly after receiving ethical approval and prior to the start of recruitment, the country went into its first lockdown to prevent onward transmission of the virus. Initially, the study was paused as we knew that participant recruitment through primary and secondary care would be impacted. As time progressed, however, we decided to adapt the study to take account of the situation and the change in operating practices across sites. The initial protocol included processes for face-to-face recruitment in primary care (through routine anticoagulation clinics) and in secondary care (through identification of AF bleeding cases through medical records and approach of patients on the wards).

To minimise any potential COVID-19 risk to both research staff and patients, it was agreed therefore that the study design should operate in line with current guidelines set out by the UK and Welsh Government for COVID-19 management. Up-to-date government guidance was, therefore, followed throughout the study. This had a direct consequence on some aspects of study conduct. Study visits were, therefore, changed to ensure that they could be conducted remotely, until such time where face-to-face visits were permitted. Key members of the study team monitored relevant guidance for GP surgeries and hospitals closely and necessary adjustments to study conduct were made accordingly, and consistently with any new information and/or restrictions as it was released. Where study tasks were conducted remotely, Health Research Authority advice released for conducting research during the COVID-19 pandemic was followed, where applicable. Any study visits conducted in person took place with all required precautions in place to help prevent the spread of the virus.

### Patient and public involvement

We recruited two patient and public involvement (PPI) representatives prior to initiating the study. They had an equal voice on our study steering committees and attended all meetings. They provided input on study design, selection of appropriate data collection methods/tools, development of all patient facing documents and the ethics application. Our PPIs were selected based on appropriate relatedness to the condition, that is, living with AF and prescribed anticoagulants and/or those involved with AF support groups. Our PPIs reviewed this manuscript prior to submission and are coauthors on the publication.

### RESULTS

We received initial expressions of interest from 85 participants in primary care. These expressions of interest were, however, not translated into participation in most cases (see figure 1). In addition, 18 participants started the survey on the AF forum. A total of 32 participants consented to participate in the study—14 from primary care and 18 through the AF forum. No participants were recruited through secondary care.

COVID-19 had a major impact on the study, particularly in secondary care. We had difficulty recruiting in secondary care for three reasons. First, many research nurses were transferred back to help manage front-line National Health Service activities; second, those who were still active had difficulty gaining access to participants and wards during lockdown; and thirdly, Welsh Government mandated that research nurse activity should be prioritised to COVID-19 vaccine studies. As a result, we focused on the two other recruitment routes, primary care and the AF forum.

Despite 32 participants consenting to participate, only 26 initial surveys were completed (14 from primary care and 12 from the AF forum), with 6 from the AF forum being abandoned prior to completion of any questions on REDCap. We contacted primary care contacts at least twice if we had received an expression of interest from them.

We received follow-up surveys from 11 participants (8 primary care and 3 AF forum). Figure 2 illustrates the numbers of participants expressing an interest and subsequently consenting to participate in the study.

Table 1 illustrates the responses to the general questions from the 26 participants in the primary care and AF forum groups. We only had demographic information for the primary care group, where clinical data were cross referenced to patient records. The primary care patients were all white and mostly female (79%). The HAS-BLED score (all less than 3) indicated that participants were at low risk of bleeding.

In both the primary care and AF forum groups, participants reported many pre-existing conditions. The most common were hypertension (reported by 43% of participants) and other (46%). There were a range of times since the bleed, including beyond 30 days, despite recruitment aiming to focus on bleeds of 30 days or less. Minor bleeding was more common than major bleeding in both groups, with 6/14 (42.9%) of participants in the primary care group and 5/12 (41.7%) in the AF forum group indicating that they had fairly frequent minor bleeds (1–2 per week or 1–2 per month). This was in contrast to

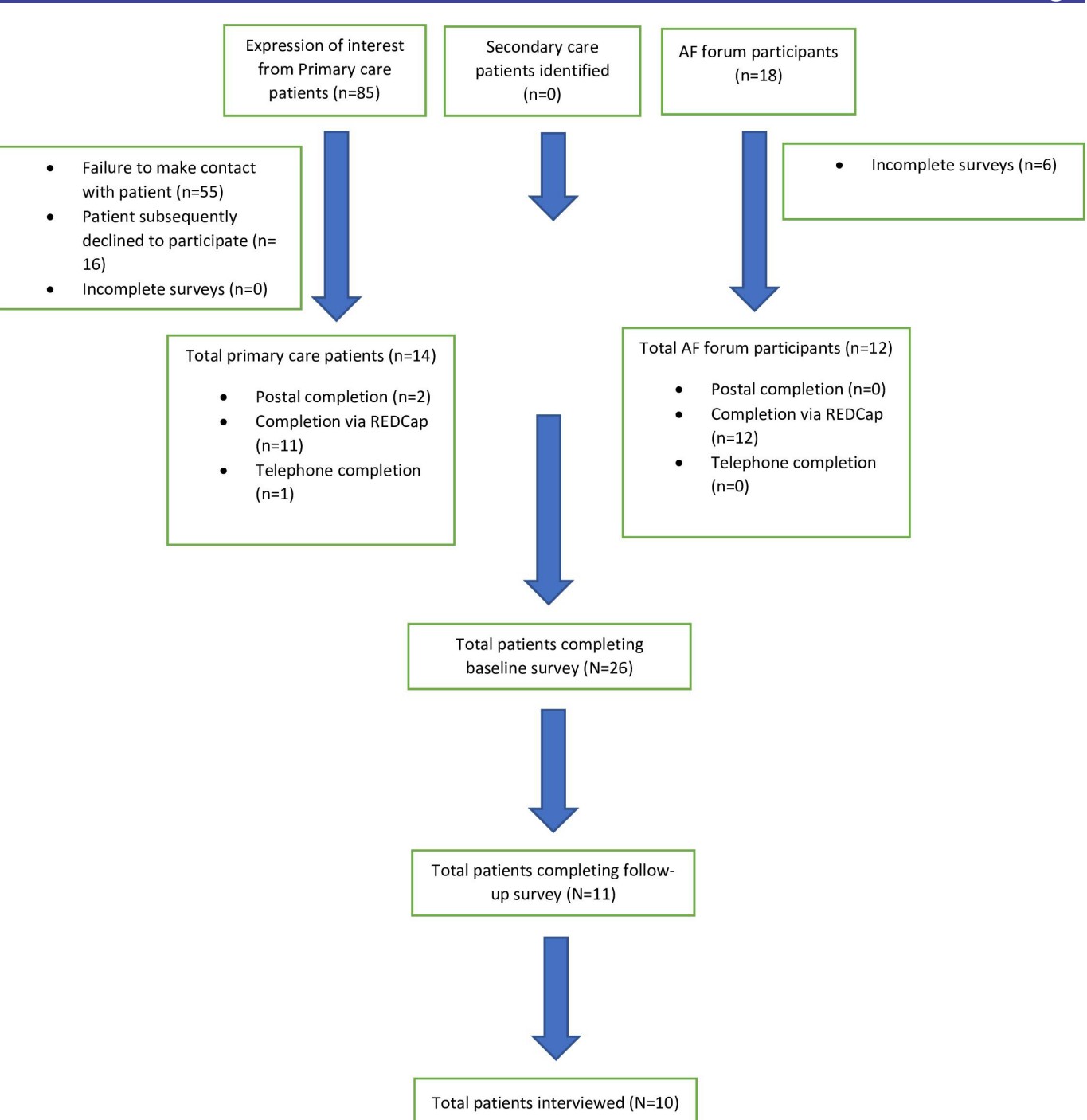

**Figure 2** Flow chart of the participants recruited to the study and subsequently analysed.

the frequency of major bleeds, where 10/14 (71.4%) of participants from primary care and 7/12 (58.3%) from the AF forum indicated that they almost never had major bleeds. Clinical classification of the most recent bleed (where documented) indicated that there were 18 minor, 2 major and 1 unknown.

Table 2 illustrates the HRQoL scores from the three questionnaires. The mean EQ-5D UK crosswalk and EQ-5D VAS scores in the primary care group were lower than that in the AF forum group at baseline (0.470 vs 0.809 crosswalk; 58.9 vs 70.6 VAS). The follow-up scores

for both the cross walk and VAS scores in the primary care group were higher, but still less than the AF forum group (0.694 vs 0.893 crosswalk; 60.1 vs 83.0 VAS).

The AFEQT overall and subscale scores were largely in the positive range indicating limited disability due to AF. At baseline, the global AFEQT score, symptom, daily activities and treatment concern domain scores in the primary care and AF forum group were similar for the global, symptom and daily activities scores (64.8 vs 65.7 global; 76.4 vs 78.0 symptom; 50.0 vs 58.7 daily activities). The participants in the primary care group had

**Table 1** Demographics of the recruited participants

| | Primary care (n=14) | AF forum (n=12)* |
|---|---|---|
| **Ethnicity** | | |
| White | 14 (100%) | n/a |
| **Sex (M:F)** | 3:11 | n/a |
| **CHA$_2$DS$_2$-VASc** | | |
| Mean (min, max) | 3.64 (1, 6) | n/a |
| **HAS-BLED** | | |
| 0 | 1 (7.1%) | n/a |
| 1 | 5 (35.7%) | |
| 2 | 8 (57.1%) | |
| **Medication** | | |
| Apixaban | 7 (50.0%) | 1 (8.3%) |
| Dabigatran | 0 (0%) | 1 (8.3%) |
| Rivaroxaban | 5 (35.7%) | 4 (33.3%) |
| Warfarin | 2 (14.3%) | 2 (16.7%) |
| Xarelto | 0 (0%) | 1 (8.3%) |
| Missing | 1 (7.1%) | 3 (25.0%) |
| **Comorbidities/pre-existing disease** | | |
| Congestive heart failure | 0 (0%) | 1 (8.3%) |
| Hypertension | 6 (42.9%) | 5 (41.7%) |
| Hypotension | 1 (7.1%) | 0 (0%) |
| Ischaemic heart disease | 0 (0%) | 1 (8.3%) |
| Chronic obstructive pulmonary disease | 2 (14.3%) | 0 (0%) |
| Diabetes related heart disease | 1 (7.1%) | 1 (8.3%) |
| Other | 9 (64.3%) | 3 (25.0%) |
| **Days since bleed** | | |
| 0–7 | 3 (21.4%) | 2 (16.7%) |
| 8–30 | 0 (0%) | 2 (16.7) |
| 31–60 | 1 (7.1%) | 4 (33.3%) |
| 60 | 1 (7.1%) | 0 (0%) |
| Unknown | 9 (64.3%) | 4 (33.3%) |
| **Location of bleed** | | |
| Internal | 3 (21.4%) | 4 (33.3%) |
| External | 7 (50%) | 4((33.3%) |
| Unsure | 1 (7.1%) | 1 (8.3%) |
| Missing | 3 (21.4%) | 3 (25.0%) |
| **Body location of external bleed** | | |
| Head | 0 (0%) | 2 (50.0%) |
| Neck | 1 (14.3%) | 0 (0%) |
| Forearm | 2 (28.6%) | 0 (0%) |
| Calf/shin | 1 (14.3%) | 1 (25.0%) |

Continued

**Table 1** Continued

| | Primary care (n=14) | AF forum (n=12)* |
|---|---|---|
| Feet, ankle, toes | 1 (14.3%) | 0 (0%) |
| Haemorrhoids | 1 (14.3%) | 0 (0%) |
| Missing | 1 (14.3%) | 1 (25.0%) |
| **Body location of internal bleed** | | |
| Head | 2 (66.7%) | 0 (0%) |
| Respiratory system | 0 (0%) | 1 (25.0%) |
| Reproductive system | 0 (0%) | 1 (25.0%) |
| Other | 1 (33.3%) | 2 (50.0%) |
| **Frequency of minor bleeds** | | |
| 1–2 times per week | 4 (28.6%) | 1 (8.3%) |
| 1–2 times per month | 2 (14.3%) | 4 (33.3%) |
| 1–2 times over 6 months | 3 (21.4%) | 2 (16.7%) |
| 1–2 times a year | 0 (0%) | 1 (8.3%) |
| Almost never | 3 (21.4%) | 2 (16.7%) |
| Missing | 2 (14.3%) | 2 (16.7%) |
| **Frequency of major bleeds** | | |
| 1–2 per week | 1 (7.1%) | 0 (0%) |
| 1–2 times a year | 0 (0%) | 2 (16.7%) |
| Almost never | 10 (71.4%) | 7 (58.3%) |
| Missing | 3 (21.4%) | 3 (25.0%) |

HAS-BLED score indicates risk of major bleeding. Calculated based on: history of hypertension, liver disease, renal disease, stroke history, prior major bleeding or predisposition to bleeding; labile International Normalised Ratio (INR) results; age; medication usage predisposing to bleeding; alcohol use. CHA$_2$DS$_2$VASc score indicates AF stroke risk. Calculated based on: age; sex; congestive heart failure risk; hypertension history; stroke/TIA/thromboembolism history; vascular disease history (prior myocardial infarction, peripheral artery disease or aortic plaque); and diabetes history.
*Demographic/symptom information was not collected on the AF forum as it could not be clinically verified.
AF, atrial fibrillation; n/a, not available.

less treatment concern at baseline that the AF forum group (76.9 vs 66.9; treatment concern). The global and symptom scores showed little change between the baseline and follow-up survey in the primary care group (64.8 vs 63.3 global; 76.4 vs 74.0). The primary care group score did however improve for daily activities (50.0 vs 59.8; daily activities) and deteriorate for treatment concern (76.9 vs 59.1). In the AF forum group, there was an improvement in the global score (65.7 vs 77.2 global) and little change in the symptom score (78.0 vs 81.9 symptom). There was also an improvement in the daily activities (58.7 vs 76.4 daily activities) and treatment concern (66.9 vs 75.0) score in the AF group between baseline and follow-up.

**Table 2** Health-related quality of life scores at baseline and follow-up split by primary care and AF forum patients measured using the EQ-5D, AFEQT and PACT questionnaires

**(A) EQ-5D UK crosswalk scores**

**EQ-5D UK crosswalk***

| Mean (min, max) | First survey | | Follow-up survey | | Total | |
|---|---|---|---|---|---|---|
| Primary care | n=13 | 0.470 (−0.200, 1.000) | n=8 | 0.694 (0.393, 1.000) | n=21 | 0.555 (−0.200, 1.000) |
| Online forum | n=12 | 0.809 (0.476, 1.000) | n=3 | 0.893 (0.679, 1.000) | n=15 | 0.826 (0.476, 1.000) |
| Total | n=25 | 0.633 (−0.200, 1.000) | n=11 | 0.748 (0.393, 1.000) | n=36 | 0.668 (−0.200, 1.000) |

**EQ-5D UK VAS**

| Mean (min, max) | First survey | | Follow-up survey | | Total | |
|---|---|---|---|---|---|---|
| Primary care | n=13 | 58.9 (22, 92) | n=8 | 60.1 (25, 98) | n=21 | 59.4 (22, 98) |
| Online forum | n=12 | 70.6 (25, 92) | n=3 | 83.0 (70, 90) | n=15 | 73.1 (25, 92) |
| Total | n=25 | 64.5 (22, 92) | n=11 | 66.4 (25, 98) | n=36 | 65.1 (22, 98) |

**(B) AFEQT scores**

**AFEQT**

| Mean (min, max) | First survey | Follow-up survey |
|---|---|---|
| Global score | | |
| Primary care | 64.8 (18.5, 98.1) | 63.3 (37.0, 91.7) |
| AF forum | 65.7 (25.9, 94.4) | 77.2 (50.5, 93.5) |
| Symptom Domain Score | | |
| Primary care | 76.4 (33.3, 100) | 74.0 (37.5, 95.8) |
| AF forum | 78.0 (37.5, 100) | 81.9 (58.3, 95.8) |
| Daily Activities Domain Score | | |
| Primary care | 50.0 (14.6, 100) | 59.8 (8.3, 95.8) |
| AF forum | 58.7 (10.4, 91.7) | 76.4 (47.9, 95.8) |
| Treatment Concern Domain Score | | |
| Primary care | 76.9 (13.9, 100) | 59.1 (11.1, 83.3) |
| AF forum | 66.9 (33.3, 91.4) | 75.0 (50.0, 91.7) |

**AFEQT satisfaction questions**

| Mean (min, max) | First Survey | Follow-up Survey |
|---|---|---|
| Q19. How well your current treatment controls your atrial fibrillation? | | |
| Primary care | 75.0 (50.0, 100) | 62.5 (16.7, 100) |
| AF forum | 65.3 (0.0, 83.3) | 61.1 (50.0, 66.7) |
| Q20. The extent to which treatment has relieved your symptoms of atrial fibrillation? | | |
| Primary care | 73.6 (50.0, 100) | 62.5 (16.7, 83.3) |
| AF forum | 66.7 (0.0, 100) | 61.1 (50.0, 66.7) |

**(C) PACT-Q Part 2 Scores**

**PACT-Q scores**

| Mean (min, max) | First survey | Follow-up survey |
|---|---|---|
| Convenience score | | |
| Primary care | 87.0 (59.6, 100) | 86.8 (69.2, 100) |
| AF forum | 86.5 (65.4, 98.1) | 86.5 (76.9, 98.1) |
| Satisfaction score | | |
| Primary care | 65.4 (39.3, 96.4) | 61.6 (50.0, 78.6) |
| AF forum | 67.5 (39.3, 89.3) | 76.2 (64.3, 92.9) |

*The crosswalk has a score range of −0.594 to 1.000.
AF, atrial fibrillation; AFEQT, atrial fibrillation effect on quality of life; EQ5D, EuroQol 5 dimensions; PACT-Q, Perception of Anticoagulant Treatment Questionnaire; VAS, Visual Analogue Scale.

Both groups had a deterioration in response to the two AFEQT satisfaction questions (Q19 and 20, see table 2). These questions relate to how well they think their current treatment controls their AF and the extent to which their treatment has relieved their symptoms.

The PACT-Q2 convenience scores were comparable between the primary care and AF forum groups at baseline (87.0 vs 86.5 convenience). There was little change in these scores at follow-up in the primary care (87.0 vs 86.8) or the AF forum group (86.5 vs 86.5). The initial PACT-Q2 satisfaction scores were again comparable between the primary care and AF forum group (65.4 vs 67.5 satisfaction). The primary care group however showed a deterioration in the satisfaction score at follow-up (65.4 vs 61.6), whereas the AF forum group showed an improvement in scores (67.5 vs 76.2) at follow-up.

Due to the small numbers participating in the surveys, it is difficult to infer any meaningful conclusions from these results and as such they should be interpreted with caution.

## DISCUSSION

After receiving the necessary ethics and research and development (R&D) approvals for the study, the UK went into lockdown, and we were forced to initially pause the study. When we were in a position to restart, we had to modify the study to take account of changing healthcare practices. This included remote engagement and varying formats for completion of the questionnaires. We used various approaches to increase recruitment, including the option to complete the survey on paper, through an online survey link or over the phone. Despite receiving many expressions of interest and making multiple phone calls, we were unable to make contact with most patients who had expressed an interest in participating. The study was open for a year longer than planned (2 years in total) after which the research management group made the decision to close.

Recruitment in secondary care was mostly impacted and we were unable to recruit any patients through this route. This was largely due to nurses being reassigned to frontline roles, limited access to wards and a focus on recruitment to COVID-19 studies. We were still able to recruit through primary care and the AF online forum. There were, however, a number of incomplete questionnaires submitted through the online forum. This may indicate that patients find online forms difficult to complete[55] or that they were unsure of the relevance of the questions or found the survey too long. Separate qualitative interviews with patients will explore these concepts further.

Primary care, therefore, appeared to be the best way to recruit participants. This may have been because they were introduced to the study by the primary care pharmacist whom they knew, and they were therefore not contacted 'cold'. We received initial expressions of interest in primary care from 85 participants. This did not, however, translate into 85 completed questionnaires. This may be because of the time lapse between discussing the study and subsequently being contacted by the researcher or misunderstanding regarding the rationale for the study. Had the participants received the questionnaire on initial contact, this may have increased the completion rate.

Most of the patients who completed the survey had minor bleeds. This may be due to the fact that most patients were recruited through primary care. Had we recruited through secondary care, we may have seen more major bleeds.

Selecting the PROMs for the study was challenging. The focus of the study was to test the feasibility of collecting data regarding HRQoL in AF participants on OATs who experienced a bleed. Details of the selection process for the PROMs will be reported separately, but the final decision was based on (1) previous use of the PROMs with the AF population; (2) assessment of the validity of the PROMs and (3) relevance of the questions to a bleeding event. There were few PROMs that had questions that specifically focused on the bleed. This may have also contributed to the fact that the response rate was lower than expected.

It was difficult to interpret the HRQoL scores as the study was not powered to detect any statistically significant differences. In addition, there was a great deal of variability in the scores across the different scales even within the same groups. Due to the limited number of participants recruited and the variability in the scores obtained, contextualising these scores with existing literature is therefore difficult. Further studies should explore whether the type of OAT has any impact on HRQoL, particularly as some OATs are less likely to cause bleeding. We collected information regarding existing comorbidities and a future study should also explore whether any of these conditions are likely to affect the risk of bleeding.

Further work needs to be undertaken to explore alternative ways of recruiting through secondary care. In our study, we used research nurses with the aim of accessing ward admission records to identify potential patients. This mechanism for recruitment was particularly impacted by the pandemic; access to the wards was restricted, many research nurses were moved back to front-line activities, and COVID-19 research studies were prioritised. Recruitment through secondary care may be improved by working directly with clinicians who manage AF patients on OATs. It is likely that this would also increase the number of participants recruited with major bleeds and help to determine the most appropriate time to approach participants, especially as they may be too ill during their inpatient episode.

The way we recruited through primary care could also be streamlined, with participants being invited to participate following their initial contact with the lead clinician managing the bleeding clinics rather than being contacted by the researcher following completion of an expression of interest. The relevance of the questionnaires may have impacted on the completion of the survey through

the online forum, so use of a more specific PROM that focused on bleeding issues may facilitate this.

Despite the challenges of undertaking the study in the pandemic, we were still able to recruit 32 patients. A further fully powered study is now needed to explore whether HRQoL is impacted by bleeding episodes in AF patients on OATs, and whether the extent of the bleed has an effect on HRQoL.

**Author affiliations**
[1]Swansea University Medical School, Swansea University, Swansea, UK
[2]Department of Haematology, University Hospital of Wales, Cardiff, UK
[3]Swansea City GP Cluster, Swansea, UK
[4]VBHC Academy, School of Management, Swansea University, Swansea, UK
[5]Swansea University, Swansea, UK
[6]Arrythmia Alliance, Swansea, UK
[7]Bristol-Myers Squibb Pharmaceuticals Ltd, Uxbridge, UK

**Acknowledgements** We would like to thank all participants for completing the survey, the general practice nurses from Swansea Bay City Cluster who helped with recruitment and the Arrhythmia Alliance for hosting the online survey.

**Contributors** HAH and SL conceived the study. HAH designed the initial study proposal, led on interpretation of the findings and drafted the manuscript. KJL was responsible for site setup, ethical approvals, participant recruitment and data collection; participated in interpretation of findings and provided input on the manuscript. GH provided overall study governance support, specifically in relation to ethical aspects, provided input into final protocol, participated in interpretation of findings and provided input on the manuscript. RA provided input into final protocol, was responsible for clinical interpretation of bleeds, participated in interpretation of findings and provided input on the manuscript. RJ led primary care recruitment, provided input into final protocol, participated in interpretation of findings and provided input on the manuscript. TL led AF forum recruitment, provided input into final protocol, provided PPI input on survey design, participated in interpretation of findings and was provided input on the manuscript. AH provided input into final protocol, provided PPI input on survey design, participated in interpretation of findings and provided input on the manuscript. DT was responsible for writing the analysis plan, final analysis and interpretation of the findings and provided input on the manuscript. HL and KP provided input into final protocol, participated in interpretation of findings and provided input on the manuscript. All authors reviewed and approved the final manuscript. HAH is the guarantor for the manuscript and accepts full responsibility for the work and/or the conduct of the study, had access to the data and controlled the decision to publish.

**Funding** The chief investigator of the study was HAH, Swansea Trials Unit, Swansea University (SU). Under the UK Policy Framework for Health and Social Care Research SU met the requirements of research governance sponsor. BMS/Pfizer funded the research (grant number: CV185-770) and under the UK policy framework were part of a partnership that accepted overall responsibility for proportionate and effective arrangements to be in place to set up, run and report the research project. They were represented on the Trial Steering Committee throughout the project.

**Competing interests** HAH leads this independent research which was funded by Pfizer and Bristol Myers Squibb. HL is an employee of Swansea University who has received a grant for a Value-Based Healthcare programme of work from Pfize, which he leads and support to attend a meeting. SL was an employee of BMS at the time of the study and holds stock in BMS and Pfizer. KP is an employee of BMS and holds stock in BMS. GH, KJL, AH, RJ and TL, declare that they have no conflict of interest.

**Patient and public involvement** Patients and/or the public were involved in the design, or conduct, or reporting, or dissemination plans of this research. Refer to the Methods section for further details.

**Patient consent for publication** Not applicable.

**Ethics approval** We complied with Good Clinical Practice (GCP) guidelines and the General Data Protection Regulation (GDPR) in the conduct of the study and the collection of participant information. The study was given ethical approval from the London - Riverside Research Ethics Committee Research Ethics Committee (IRAS ID 279646), and Swansea Bay University Health Board R&D. The study research governance sponsor was Swansea University.

**Provenance and peer review** Not commissioned; externally peer reviewed.

**Data availability statement** Data are available in a public, open access repository. The anonymised dataset will be made available on the Zenodo open access data repository.

**ORCID iDs**
Hayley Anne Hutchings http://orcid.org/0000-0003-4155-1741
Kirsty J Lanyon http://orcid.org/0000-0002-4227-6852
Hamish Laing http://orcid.org/0000-0002-5661-7937

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
