## [Reviewer comments · BMJ Open]

ARTICLE DETAILS

TITLE (PROVISIONAL)	Can we collect health-related quality of life information from anticoagulated atrial fibrillation participants who have recently experienced a bleed? An observational feasibility study in primary, and secondary care in Wales and through a UK online forum
AUTHORS	Hutchings, Hayley; Lanyon, Kirsty; Holland, Gail; Alikhan, Raza; Jenkins, Rhys; LAING, HAMISH; Hughes, Arfon; Lobban, Trudie; Pollock, Kevin; Tod, Daniel; Lister, Steven

VERSION 1 – REVIEW

REVIEWER	Lomper, Katarzyna Medical University, Poland
REVIEW RETURNED	15-Jun-2023

GENERAL COMMENTS	1 First of all, the study group is too small for the population of patients with atrial fibrillation and bleeding. Perhaps the study period should be extended? The authors suggest limitations of the study as a result of the Covid-19 pandemic, however, in the methodology they indicate that the questionnaires were completed by patients in person or during a telephone interview. Thus, if the surveys were conducted over the phone, a larger group of respondents could be obtained. 2 A description of the tools used should be in the survey methodology section, not next to the results. 3. in the study, information about the type of anticoagulants taken by the patients can be found only in the results section - please describe in detail in the methodology which drugs were taken by the surveyed patients, in addition I suggest dividing the study into VKA and NOAC, because NOAC is much less likely to cause bleeding - it would be good to examine the relationship between the type of drug taken and the examined variables. 4. I suggest using the nomenclature HRQOL, not QoL 5. did the patients have comorbidities that could affect the risk of bleeding? 6. the study did not examine the relationship between the results of, for example, paco2 and bleeding, and it may increase the value of the paper.
--

REVIEWER	C. Pierre-Louis, Isabelle UMass Chan Medical School, Department of Population and Quantitative Health Sciences
REVIEW RETURNED	23-Jun-2023

GENERAL COMMENTS	Nice paper. I do think the results section could benefit from description of results as opposed "please see figure xx", specifically with the QoL answers. Table 1 can also be condensed,
---

	i.e. give the average (SD) for the CHAD score. It was difficult to understand table 2a. In the methods, the EQ5D is scored between 0 and 100, so I am unsure of how to read or interpret the numbers in the table.
--	--

VERSION 1 – AUTHOR RESPONSE

Reviewer: 1

Dr. Katarzyna Lomper, Medical University, Poland

Comments to the Author:

1 First of all, the study group is too small for the population of patients with atrial fibrillation and bleeding. Perhaps the study period should be extended? The authors suggest limitations of the study as a result of the Covid-19 pandemic, however, in the methodology they indicate that the questionnaires were completed by patients in person or during a telephone interview. Thus, if the surveys were conducted over the phone, a larger group of respondents could be obtained.

Response: We agree with the reviewer's observations and suggestions. We did however try various approaches to increase recruitment, including multiple phone calls, but were unable to make contact with most patients who had expressed an interest in participating. The study was open for a year longer than planned (2 years in total) after which the research management group made the decision to close. We have added this information into the manuscript.

2 A description of the tools used should be in the survey methodology section, not next to the results.

Response: Thank you. We have now moved the description of the tools from the results to the survey methodology section.

3. in the study, information about the type of anticoagulants taken by the patients can be found only in the results section - please describe in detail in the methodology which drugs were taken by the surveyed patients, in addition I suggest dividing the study into VKA and NOAC, because NOAC is much less likely to cause bleeding - it would be good to examine the relationship between the type of drug taken and the examined variables.

Response: Thank you. We have added information in the methodology section to provide more detail regarding medication use. We agree that it would be useful in a future study to explore the impact of different types of oral anticoagulants on quality of life, especially as some are less likely to cause bleeding. As a feasibility study however, we focussed on processes at this stage rather than examining differences across groups. We have added in the discussion that a future study should explore possible differences relating to drug use.

4. I suggest using the nomenclature HRQOL, not QoL

Response: We have now changed QoL to HRQoL throughout.

5. did the patients have comorbidities that could affect the risk of bleeding?

Response: We collected information regarding co-morbidities which are documented in Table 1. As the study was not powered to detect statistically significant differences, we have not explored the impact of existing co-morbidities on HRQoL. We have however added a sentence in the discussion to suggest that future work should explore the impact of existing co-morbidities.

6. the study did not examine the relationship between the results of, for example, pactq and bleeding, and it may increase the value of the paper

Response: The purpose of the study was to explore feasibility of undertaking a fully powered study. We agree that it would be useful to explore this, but the sample size in the current study would preclude us from making any meaningful conclusions.

Reviewer: 2

Dr. Isabelle C. Pierre-Louis, UMass Chan Medical School

Comments to the Author:

Nice paper. I do think the results section could benefit from description of results as opposed "please see figure xx", specifically with the QoL answers.

Response: Thank you for this feedback. We have now included more description of the findings in the results section as well as in the tables.

Table 1 can also be condensed, i.e. give the average (SD) for the CHAD score.

Response: We have now condensed Table 1 and include the average CHAD score.

It was difficult to understand table 2a. In the methods, the EQ5D is scored between 0 and 100, so I am unsure of how to read or interpret the numbers in the table.

Response: Apologies for this confusion. We have now provided more detail in the methodology section regarding the two components of the EQ-5D scores and have presented these findings in the results and tables.

Reviewer: 1

Competing interests of Reviewer: I have nothing to declare

Reviewer: 2

Competing interests of Reviewer: None.